# Unsupervised Object-Level Representation Learning from Scene Images

**Jiahao Xie**[1]   **Xiaohang Zhan**[2]   **Ziwei Liu**[1]   **Yew Soon Ong**[1,3]   **Chen Change Loy**[1]

[1]Nanyang Technological University
[2]The Chinese University of Hong Kong
[3]A∗STAR, Singapore
`{jiahao003, ziwei.liu, asysong, ccloy}@ntu.edu.sg`
`xiaohangzhan@outlook.com`

## Abstract

Contrastive self-supervised learning has largely narrowed the gap to supervised pre-training on ImageNet. However, its success highly relies on the object-centric priors of ImageNet, *i.e.*, different augmented views of the same image correspond to the same object. Such a heavily curated constraint becomes immediately infeasible when pre-trained on more complex scene images with many objects. To overcome this limitation, we introduce **O**bject-level **R**epresentation **L**earning (ORL), a new self-supervised learning framework towards scene images. Our key insight is to leverage image-level self-supervised pre-training as the prior to discover object-level semantic correspondence, thus realizing object-level representation learning from scene images. Extensive experiments on COCO show that ORL significantly improves the performance of self-supervised learning on scene images, even surpassing supervised ImageNet pre-training on several downstream tasks. Furthermore, ORL improves the downstream performance when more unlabeled scene images are available, demonstrating its great potential of harnessing unlabeled data in the wild. We hope our approach can motivate future research on more general-purpose unsupervised representation learning from scene data.[1]

## 1   Introduction

Unsupervised visual representation learning aims at obtaining transferable features with abundant unlabeled data. Recent self-supervised learning (SSL) methods based on contrastive learning [60, 22, 37, 5, 19, 4, 7] have largely narrowed the gap and even surpassed the supervised counterpart on a number of downstream tasks [30, 49, 15, 47, 35, 23]. These methods build upon the instance discrimination task that maximizes the agreement between different data-augmented views of the same image. Despite their success, current SSL methods are primarily pre-trained on the unlabeled ImageNet [8] dataset that contains iconic images with single object as shown in Figure 1(a). The underlying object-centric constraint of ImageNet makes it hard to be applied in real world scenarios where more complex scene images with multiple objects are available. Meanwhile, naïvely adopting the off-the-shelf contrastive learning methods on scene images introduces inconsistent learning signals since random crops of the same image may correspond to different objects as shown in Figure 1(b). Indeed, it has been shown that current contrastive learning methods tend to struggle on more complex scene datasets [19, 50, 34, 58] like COCO [33] or Places365 [72]. Therefore, it is imperative to design an effective object-level representation learning paradigm as illustrated in Figure 1(c) to harness massive unlabeled scene images in the wild.

---

[1]Project page: `https://www.mmlab-ntu.com/project/orl/`.

35th Conference on Neural Information Processing Systems (NeurIPS 2021).

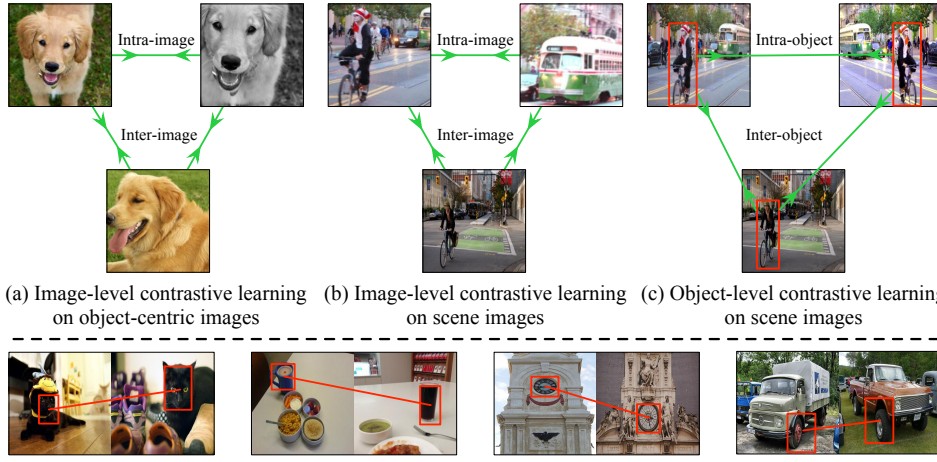

(a) Image-level contrastive learning on object-centric images

(b) Image-level contrastive learning on scene images

(c) Object-level contrastive learning on scene images

(d) Example region correspondence discovered across images

Figure 1: (a) Current image-level contrastive learning methods heavily rely on the object-centric bias of ImageNet, *i.e.*, different crops correspond to the same object. Prior works use either the different views of the same image [60, 22, 37, 5, 19, 7] (*i.e.*, intra-image) or similar images [74, 63, 1, 11] (*i.e.*, inter-image) to form positive pairs. (b) Directly adopting image-level contrastive learning methods on scene images can cause inconsistent learning signals since different crops may correspond to different objects. (c) Object-level contrastive learning can overcome the limitation in (b) by enforcing object-level consistency. (d) We find that image-level contrastive learning encodes priors for region correspondence discovery across images, and high-response regions are usually objects or object parts (we show one discovered object-instance pair per image pair for clarity), which is useful for object-level representation learning.

In this work, we are interested in going beyond ImageNet to obtain better representations on non-iconic images. Apparently, it is challenging to learn representations from scene-level images since they are entangled with many concepts including structures, objects, backgrounds and relationships. It remains an open question how to take advantages of spatial information of multiple objects naturally residing in the scene images when no object annotations are available, let alone further deriving object-level correspondence to construct positive object-instance pairs.

To tackle these challenges, we introduce a novel object-level unsupervised representation learning framework tailored for scene images. Our framework is based on a key insight of the current contrastive learning methods: they can implicitly group different images with similar visual concepts together even though they are explicitly optimized to group different views of the same image. This phenomenon reveals that image-level contrastive learning has already induced a latent space with rich visual concepts. Though the latent space usually entangles other scene concepts like structures, backgrounds and relationships, it will be useful for object discovery if appropriately deployed. Through computing the similarity of sampled regions between $k$-nearest-neighbor (KNN) images, we conclude two observations: 1) image-level contrastive learning encodes priors for region correspondence discovery across images; 2) high-response regions are usually objects or object parts.

Based on the observation above, we propose a multi-stage framework for unsupervised object-level representation learning. Specifically, we first extract potential object-based regions in scene images using the unsupervised region proposal algorithms (*e.g.*, selective search [56]). We then propose a region correspondence generation scheme to leverage the off-the-shelf image-level contrastive learning pre-trained model to discover corresponding object-instance pairs for the proposed regions in the embedding space. Finally, we use the obtained object-instance pairs to construct positive sample pairs for object-level representation learning. Figure 1(d) shows several cross-image object-instance pairs discovered by our framework on COCO dataset using the latent prior of BYOL [19], the state-of-the-art image-level contrastive learning method. The discovered inter-corresponding pairs substantially provide diverse intra-class variances at the object-instance level to aid object-level representation learning.

Overall, our main contributions are summarized as follows:

**1)** We observe that existing image-level contrastive learning methods have priors to discover object-level correspondence across images. We leverage this prior for the first time for unsupervised cross-image object-level correspondence discovery.

**2)** With the obtained correspondence, we introduce a novel multi-stage self-supervised learning pipeline, termed as ORL, for object-level representation learning from scene images, going beyond object-centric ImageNet.

**3)** We contribute the first study for object-level SSL. ORL substantially outperforms image-level contrastive learning approaches pre-trained on COCO dataset (∼118k images with labels discarded), setting a new state of the art on this challenging dataset that contains diverse scenes in the wild. The COCO pre-trained ORL even surpasses supervised ImageNet pre-training on several considered downstream tasks. When SSL is conducted on a larger "COCO+" dataset (COCO `train2017` set plus COCO `unlabeled2017` set, ∼241k images in total), ORL further improves the performance, demonstrating its potential to benefit from more unlabeled scene data.

## 2 Related work

**Self-supervised learning.** Self-supervised learning builds unsupervised representations by exploiting the internal priors or structures of data in the form of a pretext task. A wide range of pretext tasks have been proposed in the past few years. Examples include patch context prediction [10], jigsaw puzzles [39], inpainting [43], colorization [31, 70], cross-channel prediction [71], visual primitive counting [40], and rotation prediction [14]. Although good representations emerge with these pretext tasks, they are prone to lose generality due to their hand-crafted nature.

Recently, contrastive learning [20] that performs instance discrimination [60, 22, 37, 5, 19, 4, 7] has shown great potential in this field, largely narrowing the gap to fully supervised learning. The core idea of contrastive learning is to gather positive pairs and separate negative pairs in the embedding space. A positive pair is usually formed with two transformed views of the same image while the negative pairs are formed with different images. Typically, contrastive learning methods require a large number of negative samples to avoid mode collapse. These samples can be maintained within a mini-batch [42, 27, 66, 26, 2, 5], a memory bank [60, 53, 74, 37] or a queue [22, 6]. BYOL [19] and SwAV [4] further remove the necessity of involving negative pairs. BYOL directly predicts the features of one view from another view, while SwAV predicts the cluster assignments between multiple views of the same image. Despite their improved performance, the existing image-level contrastive learning methods are largely confined to the underlying object-centric bias of ImageNet.

More recently, a group of works that perform pixel-level [45, 58, 64, 50, 34, 25] or region-level [48, 65, 61, 62, 9] representation learning have emerged. Our work is more related to region-level representation learning but substantially different from this line of research in the following aspects: 1) they still largely pre-train on object-centric ImageNet while we pre-train on non-iconic scene images, 2) they align pre-training specifically for dense prediction downstream tasks while we target at more general-purpose representation learning that improves performance in both dense prediction and classification tasks, 3) their randomly cropped local regions do not contain the explicit *object* notion as ours, and 4) they only rely on intra-image transformations (*e.g.*, random cropping) to construct corresponding positive pairs from the same image while we leverage the discovered high-level semantic correspondence to construct positive pairs across images.

There are also a few prior attempts [18, 3, 16] for self-supervised learning on non-curated scene images. As opposed to our work, most of them consider larger models and datasets to explore the limit of current self-supervised learning methods without further considering the object-level information residing in scene images.

**Visual correspondence.** Visual correspondence aims at finding pairwise pixels or regions across images that result from the same scene [67], which can be regarded as similarity learning of visual descriptors among matched points or patches. While early efforts learn dense correspondence with labeled data [21, 68, 29, 55, 41], some recent works learn the similarity between the parts or landmarks of the data in an unsupervised manner [52, 51]. Our work substantially differs from this line of research from original intention. Previous works aim at accurately detecting all correspondence given two images, whereas our work focuses on retrieving high-quality correspondence to improve representation learning.

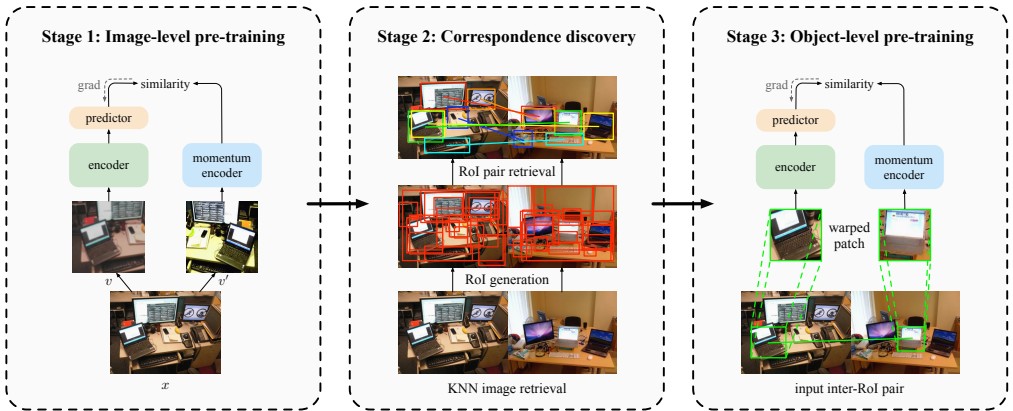

Figure 2: **Overview of our three-stage pipeline.** In Stage 1, we pre-train an image-level contrastive learning model, *e.g.*, BYOL. In Stage 2, we first use the pre-trained model to retrieve KNNs for each image in the embedding space to obtain image-level visually similar pairs. We then use unsupervised region proposal algorithms (*e.g.*, selective search) to generate rough RoIs for each image pair. Afterwards, we reuse the pre-trained model to retrieve the top-ranked RoI pairs, *i.e.*, correspondence. We find these pairs of RoIs are almost objects or object parts. In Stage 3, with the corresponding RoI pairs discovered across images, we finally perform object-level contrastive learning using the same architecture as Stage 1.

## 3 Methodology

We propose a new multi-stage self-supervised learning framework, *i.e.*, ORL, for object-level representation learning from scene images. ORL extends the existing image-level contrastive learning framework to object level by leveraging priors from image-level instance discrimination. The overall pipeline of ORL is illustrated in Figure 2. It contains three stages: image-level pre-training, correspondence discovery, and object-level pre-training. We detail each stage as follows.

### 3.1 ORL pipeline

**Preliminary: Contrastive learning.** Our pipeline contains several contrastive learning modules in Stage 1 and 3. Without loss of generality, we consider BYOL [19] as our basic contrastive learning module. BYOL uses two neural networks: the *online* network $f_\theta(x)$ and the *target* network $g_\xi(x)$. The target network provides the regression target to train the online network while its weights $\xi$ are updated by an exponential moving average of the online parameters $\theta$ with a decay rate $\tau \in [0, 1]$ following BYOL. Given two input images $x_1$ and $x_2$, the loss function is defined as:

$$\mathcal{L}(x_1, x_2) \triangleq \|f_\theta(x_1) - g_\xi(x_2)\|_2^2, \tag{1}$$

We name it an "intra-" version of BYOL if $x_1$ and $x_2$ are two augmented views from the same image, otherwise an "inter-" one.

**Stage 1: Image-level pre-training.** The foremost stage is to obtain an unsupervised pre-trained model from image-level tasks. As shown in Figure 2 Stage 1, given two augmented views $v$ and $v'$ from the same input image $x$, we pre-train the network following the loss function $\mathcal{L}_{\text{image}} = \mathcal{L}(v, v')$, constituting a standard image-level BYOL pre-training. This stage can be freely replaced with other image-level contrastive learning methods. We adopt BYOL here for its simplicity and effectiveness.

**Stage 2: Correspondence discovery.** We employ the pre-trained image-level contrastive learning model in Stage 1 to mine object-level correspondence for the whole dataset. As shown in Figure 2 Stage 2, the overall discovery process comprises three steps.

(i) Image-level nearest-neighbor retrieval. Specifically, for each query image $x$ in the training set $\mathcal{D}$, we first retrieve its top $K$ nearest neighbors $\mathcal{N}_k$, $k = 1, ..., K$, by cosine distance in the embedding space using the features learned from the first stage to form image-level pairs that contain similar visual context.

(ii) Region-of-interest (RoI) generation. To generate object-based RoIs, we apply unsupervised region proposal algorithms, *e.g.*, selective search [56], for each image in the pair. Considering the redundancy of generated proposals (each image can have thousands of proposals), we filter certain number of them with some pre-defined thresholds[2] including the minimal scale, the range of aspect ratio, and the maximal intersection-over-union (IoU) among the filtered boxes. After the filtering operation, we select the top 100 proposals ranked with objectiveness as the candidate RoI set for subsequent RoI pair retrieval. To extract features with the equally-sized input that is compatible with the backbone, we crop and resize each RoI to $224 \times 224$. Note that even top RoIs ranked with objectiveness are still very noisy, containing a large proportion of non-object regions.

(iii) Top-ranked RoI pair retrieval. For each query RoI from $x$, we compute its cosine similarity in the embedding space with all RoIs from its nearest-neighbor image $\mathcal{N}_k$ using the features learned from Stage 1 again. Within the calculated cosine similarity matrix $\mathbf{M}_k \in \mathbb{R}^{100 \times 100}$, we retrieve top-ranked $N$ RoI pairs to construct the set of object-level corresponding pairs $\{\mathcal{B}_k^n\}$, where $n = \{1, ..., N\}$. These high-response corresponding regions are almost objects or object parts. Finally, we save the nearest-neighbor image id and bounding box coordinate information of each corresponding pair.

**Stage 3: Object-level pre-training.** With the corresponding inter-image RoI (inter-RoI) pairs obtained in Stage 2, we perform object-level representation learning following the BYOL framework as shown in Figure 2 Stage 3. Specifically, given an input image $x$, we first randomly select one nearest-neighbor image $\mathcal{N}_k$ to obtain the corresponding set of inter-RoI pairs $\{\mathcal{B}_k^n\}$. We then randomly select one inter-RoI pair $\mathcal{B}_k^n$ as a positive pair. With the bounding box coordinate stored in $\mathcal{B}_k^n$, we crop the corresponding inter-RoIs from $x$ and $\mathcal{N}_k$, respectively, and resize each patch to $96 \times 96$, constituting two patches $p_1$ and $p_2$. We feed the two patches to the online network and target network separately to compute the loss $\mathcal{L}_{\text{inter-RoI}} = \mathcal{L}(p_1, p_2)$.

To make full use of discovered objects, we introduce the intra-RoI contrastive learning via augmenting object patches. Specifically, we randomly select one filtered bounding box from $x$ obtained in Stage 2, and spatially jitter the box around its original location with the following operations[3]: (i) a random box center shifting within 50% of its width and height, (ii) a random area scaling between 50% and 200% of the original box, and (iii) a random aspect ratio between $1/2$ and $2/1$. Similarly, we crop the two intra-RoIs $p$ and $p'$, and resize each patch to $96 \times 96$ for forward propagation to compute the loss $\mathcal{L}_{\text{intra-RoI}} = \mathcal{L}(p, p')$. The diverse spatial jittering of the bounding box encourages the network to preserve common object information and disregard the background, thus further improving the localization ability.

We keep the two original global views in BYOL as well since they preserve the global image-level information compared with the local patches. The final loss for our ORL can thus be formulated as:

$$\mathcal{L}_{\text{ORL}} = \lambda_1 \mathcal{L}_{\text{image}} + \lambda_2 \mathcal{L}_{\text{intra-RoI}} + \lambda_3 \mathcal{L}_{\text{inter-RoI}}, \tag{2}$$

where $\lambda_1$, $\lambda_2$, $\lambda_3$ are the loss weights to balance each term. We set all loss weights to 1 by default. Following BYOL, we also compute the symmetric loss $\widetilde{\mathcal{L}}_{\text{ORL}}$ by separately feeding $v'$, $p'$, $p_2$ to the online network and $v$, $p$, $p_1$ to the target network.

## 3.2 Implementation details

**Dataset.** We pre-train our models on the COCO `train2017` set that contains $\sim$118k images without using labels. Compared with the heavily curated object-centric ImageNet dataset, COCO contains more natural and diverse scenes in the wild, which is closer to real-world scenarios. We also perform self-supervised learning on a larger "COCO+" dataset (COCO `train2017` set plus COCO `unlabeled2017` set) to verify whether our method can benefit from more unlabeled scene data.

**Image augmentations.** The global image augmentation setting is the same as BYOL [19]: a $224 \times 224$-pixel random resized crop with a random horizontal flip, followed by a random color distortion, random grayscale conversion, random Gaussian blur and solarization. For the local patch augmentation, we directly crop the corresponding intra-RoI and inter-RoI on the input images, and

---

[2]In practice, we filter bounding boxes with the minimal scale as 96 pixels, the aspect ratio between $1/3$ and $3/1$, and the maximal IoU as 0.5.

[3]Inspired by the anchor technique used in object detection, we adopt the similar operations for our bounding box augmentations but in a continuous fashion.

resize each cropped patch to 96×96 to take place of the random resized cropping. The subsequent augmentations exactly follow the global ones.

**Network architecture.** We adopt ResNet-50 [24] as the default backbone. We use the same MLP projector and predictor as in BYOL: a linear layer with output size 4096 followed by batch normalization (BN) [28], rectified linear units (ReLU) [38], and a final linear layer with output dimension 256. We share the backbone and projector weights among the global and two local branches while the weights of predictor are not shared.

**Optimization.** For pre-training in Stage 1 and Stage 3, we use the same training hyper-parameters. Specifically, we use the SGD optimizer with a weight decay of 0.0001 and a momentum of 0.9. We adopt the cosine learning rate decay schedule [36] with a base learning rate of 0.2, linearly scaled[17] with the batch size ($lr = 0.2 \times$ BatchSize$/256$). The batch size is set to 512 by default, which is friendly to typical 8-GPU implementations. To keep the training iterations comparable with the ImageNet supervised pre-training, we train our models for 800 epochs with a warm-up period of 4 epochs. The exponential moving average parameter $\tau$ starts from 0.99 and is increased to 1 during training, following [19]. For correspondence generation in Stage 2, we retrieve top $K = 10$ nearest neighbors for each image and select top-ranked $N = 10\%$ RoI pairs for each image-level nearest-neighbor pair.

## 4 Experiments

### 4.1 Transferring to downstream tasks

We evaluate the quality of learned representations by transferring them to multiple downstream tasks. Following common protocol [18, 37], we use two evaluation setups: (i) the pre-trained network is *frozen* as a feature extractor, and (ii) the network parameters are fully *fine-tuned* as weight initialization. We provide more experimental details in the supplementary material.

| Method | Pre-train data | VOC07 mAP | ImageNet Top-1 | Places205 Top-1 | iNat. Top-1 |
|---|---|---|---|---|---|
| Random [18] | - | 9.6 | 13.7 | 16.6 | 4.8 |
| Supervised [37] | ImageNet | 87.5 | 75.9 | 51.5 | 45.4 |
| SimCLR [5] | COCO | 78.1 | 50.9 | 48.0 | 22.7 |
| MoCo v2 [6] | COCO | 82.2 | 55.1 | 48.8 | 27.8 |
| BYOL [19] | COCO | 84.5 | 57.8 | 50.5 | 29.5 |
| ORL (ours) | COCO | **86.7** | **59.0** | **52.7** | **31.8** |
| BYOL [19] | COCO+ | 87.0 | 59.6 | 52.7 | 30.9 |
| ORL (ours) | COCO+ | **88.6** | **60.7** | **54.1** | **32.0** |

Table 1: **Image classification with linear models.** All unsupervised methods are based on 800-epoch pre-training on COCO(+) with ResNet-50. We report mAP on the VOC07 dataset and top-1 center-crop accuracy on all other datasets. Numbers for all other methods are reproduced by us.

| Method | Pre-train data | VOC07 low-shot (mAP) | | | | | | | |
|---|---|---|---|---|---|---|---|---|---|
| | | 1 | 2 | 4 | 8 | 16 | 32 | 64 | 96 |
| Random | - | 9.2 | 9.4 | 11.1 | 12.3 | 14.3 | 17.4 | 21.3 | 23.8 |
| Supervied | ImageNet | 53.0 | 63.6 | 73.7 | 78.8 | 81.8 | 83.8 | 85.2 | 86.0 |
| SimCLR [5] | COCO | 33.3 | 43.5 | 52.5 | 61.1 | 66.7 | 70.5 | 73.7 | 75.0 |
| MoCo v2 [6] | COCO | 39.5 | 49.3 | 60.4 | 69.3 | 74.1 | 76.8 | 79.1 | 80.1 |
| BYOL [19] | COCO | 39.4 | 50.9 | 62.2 | 71.7 | 76.6 | 79.2 | 81.3 | 82.2 |
| ORL (ours) | COCO | **39.6** | **51.2** | **63.4** | **72.6** | **78.2** | **81.3** | **83.6** | **84.7** |
| BYOL [19] | COCO+ | 41.1 | 54.3 | 66.6 | 75.2 | 80.1 | 82.6 | 84.6 | 85.4 |
| ORL (ours) | COCO+ | **42.1** | **54.9** | **67.4** | **75.7** | **81.3** | **83.7** | **85.8** | **86.7** |

Table 2: **Low-shot image classification on VOC07** using linear SVMs trained on the fixed representations. All unsupervised methods are pre-trained on COCO(+) for 800 epochs with ResNet-50. We report mAP for each case across five runs.

**Image classification with linear models.** Following [18, 37], we assess the quality of features by training linear classifiers on top of the *fixed* representations extracted from different depths of the network for four datasets: VOC07 [12], ImageNet [8], Places205 [73], and iNaturalist18 [57]. These datasets involve diverse classification tasks ranging from object classification, scene recognition to fine-grained recognition. For VOC07, we train linear SVMs using LIBLINEAR package [13]

| Method | Pre-train data | 1% labels | | 10% labels | |
|---|---|---|---|---|---|
| | | Top-1 | Top-5 | Top-1 | Top-5 |
| Random | - | 1.6 | 5.0 | 21.8 | 44.2 |
| Supervised [69] | ImageNet | 25.4 | 48.4 | 56.4 | 80.4 |
| SimCLR [5] | COCO | 23.4 | 46.4 | 52.2 | 77.4 |
| MoCo v2 [6] | COCO | 28.2 | 54.7 | 57.1 | 81.7 |
| BYOL [19] | COCO | 28.4 | 55.9 | 58.4 | 82.7 |
| ORL (ours) | COCO | **31.0** | **58.9** | **60.5** | **84.2** |
| BYOL [19] | COCO+ | 28.3 | 56.0 | 59.4 | 83.6 |
| ORL (ours) | COCO+ | **31.8** | **60.1** | **60.9** | **84.4** |

Table 3: **Semi-supervised learning on ImageNet.** All unsupervised methods are pre-tained on COCO(+) for 800 epochs with ResNet-50. We fine-tune all models with 1% and 10% ImageNet labels, and report both top-1 and top-5 center-crop accuracy on the ImageNet validation set.

| Method | Pre-train data | COCO detection | | | COCO instance seg. | | |
|---|---|---|---|---|---|---|---|
| | | $AP^{bb}$ | $AP^{bb}_{50}$ | $AP^{bb}_{75}$ | $AP^{mk}$ | $AP^{mk}_{50}$ | $AP^{mk}_{75}$ |
| Random [54] | - | 32.8 | 50.9 | 35.3 | 29.9 | 47.9 | 32.0 |
| Supervised [54] | ImageNet | 39.7 | 59.5 | 43.3 | 35.9 | 56.6 | 38.6 |
| SimCLR [5] | COCO | 37.0 | 56.8 | 40.3 | 33.7 | 53.8 | 36.1 |
| MoCo v2 [6] | COCO | 38.5 | 58.1 | 42.1 | 34.8 | 55.3 | 37.3 |
| Self-EMD [34] | COCO | 39.3 | 60.1 | 42.8 | - | - | - |
| DenseCL [58] | COCO | 39.6 | 59.3 | 43.3 | 35.7 | 56.5 | 38.4 |
| BYOL [19] | COCO | 39.5 | 59.3 | 43.2 | 35.6 | 56.5 | 38.2 |
| ORL (ours) | COCO | **40.3** | **60.2** | **44.4** | **36.3** | **57.3** | **38.9** |
| BYOL [19] | COCO+ | 40.0 | 60.1 | 44.0 | 36.2 | 57.1 | 39.0 |
| ORL (ours) | COCO+ | **40.6** | **60.8** | **44.5** | **36.7** | **57.9** | **39.3** |

Table 4: **Object detection and instance segmentation fine-tuned on COCO.** All unsupervised methods are based on 800-epoch pre-training on COCO(+). We use Mask R-CNN R50-FPN (1× schedule), and report bounding-box AP ($AP^{bb}$) and mask AP ($AP^{mk}$). Numbers for MoCo v2 are adopted from [58].

following the setup in [18, 37]. We train on `trainval` split of VOC07 and evaluate mAP on `test` split. For ImageNet, Places205 and iNaturalist18, we follow [71, 18, 37] and train a 1000-way, 205-way and 8142-way linear classifier, respectively. We train on `train` split of each dataset, and report top-1 center-crop accuracy on the respective `val` split. Table 1 reports the results for the best-performing layer of each method. ORL substantially outperforms the BYOL baseline on all four datasets. We also observe that the COCO pre-trained ORL surpasses the supervised ImageNet pre-trained counterpart on Places205 by 1.2% in top-1 accuracy. This is the first time that a self-supervised learner outperforms the ImageNet pre-training using only ∼1/10 images compared with ImageNet. When pre-trained on a larger COCO+ dataset, ORL again outperforms BYOL. Note that apart from Places205 (2.6% gains), ORL also surpasses the supervised ImageNet counterpart on VOC07 by 1.1% mAP, using merely ∼1/5 images.

**Low-shot image classification.** We perform low-shot image classification with few training examples per class on VOC07 dataset following the same setup in [18]. We vary the number of labeled examples per category used to train linear SVMs on `train` split of VOC07 and report the average mAP across five independent samples for each low-shot case evaluated on `test` split. Table 2 provides the results. ORL shows consistent performance improvement over BYOL for each low-shot value, with larger gains achieved as the number of labeled examples per class is increasing. ORL also gradually bridges the gap to the supervised ImageNet pre-training under this scenario. We observe consistent performance boost when pre-training on the COCO+ dataset. Note that the COCO+ pre-trained ORL again outperforms the supervised ImageNet pre-training when the low-shot samples are 64 and 96.

**Semi-supervised learning.** We perform semi-supervised learning on ImageNet following the protocol of previous studies [60, 26, 37, 5, 19]. Specifically, we first randomly select 1% and 10% labeled data from ImageNet `train` split. We then fine-tune our models on these two training subsets and report both top-1 and top-5 accuracy on the official `val` split of ImageNet in Table 3. Again, ORL outperforms BYOL as well as the supervised ImageNet counterpart by large margins.

**Object detection and segmentation.** We train a Mask R-CNN model [23] with R50-FPN backbone [32] implemented in Detectron2 [59]. We *fine-tune* all layers end-to-end on COCO `train2017` split with the standard 1× schedule and evaluate on COCO `val2017` split. We follow the same setup in [54], with batch normalization layers synchronized across GPUs [44]. As shown in Table 4, ORL yields 0.8% AP and 0.7% AP improvements over BYOL for object detection and instance segmentation, respectively. The improvements are consistent over all evaluation metrics. When

Table 5: **Ablations for ORL.** (a) Effect of intra-RoI and inter-RoI losses. (b) Effect of NNs and RoI pairs. (c) Comparison with multi-crop BYOL. (d) Comparison with ground truth bounding boxes. (e) Pre-trainig schedules. We report mAP of linear SVMs on VOC07 classification benchmark.

(a)

| pre-train | intra-RoI | inter-RoI | VOC07 |
|---|---|---|---|
| BYOL | | | 84.5 |
| | $\checkmark$ | | 85.7 |
| ORL | | $\checkmark$ | 85.9 |
| | $\checkmark$ | $\checkmark$ | 86.7 |

(b)

| | # of NNs | | | # of RoI pairs | | |
|---|---|---|---|---|---|---|
| | 1 | 10 | 20 | 5% | 10% | 20% |
| VOC07 | 84.7 | 86.7 | 87.0 | 86.1 | 86.7 | 86.4 |

(c)

| pre-train | input views | VOC07 |
|---|---|---|
| BYOL | $2 \times 224 + 4 \times 96$ | 84.0 |
| ORL | $2 \times 224 + 4 \times 96$ | 86.7 |

(d)

| boxes | VOC07 |
|---|---|
| GT | 85.4 |
| SS | 86.7 |

(e)

| pre-train | 100 | 200 | 400 | 800 | 1600 |
|---|---|---|---|---|---|
| BYOL | 77.1 | 81.8 | 83.7 | 84.5 | 84.9 |
| ORL | 83.5 | 85.2 | 86.3 | 86.7 | 87.1 |

pre-trained on the COCO+ dataset, ORL again outperforms BYOL. It should be well noted that ORL even outperforms the most recent Self-EMD and DenseCL that are specifically designed for dense prediction downstream tasks. More importantly, either COCO or COCO+ pre-trained ORL can surpass the supervised ImageNet pre-training on all metrics. This further demonstrates the superiority of learning unsupervised representations at the object level.

## 4.2 Ablation study

In this subsection, we conduct extensive ablation experiments to examine the effect of each component that contributes to ORL. We pre-train our models on COCO and observe the downstream performance of all ablations on VOC07 SVM classification benchmark as introduced in Section 4.1.

**Effect of intra-RoI and inter-RoI losses.** Table 5a ablates the effect of our introduced $\mathcal{L}_{\text{intra-RoI}}$ and $\mathcal{L}_{\text{inter-RoI}}$ losses in Equation 2. Adding either $\mathcal{L}_{\text{intra-RoI}}$ or $\mathcal{L}_{\text{inter-RoI}}$ can improve the performance, with the best results obtained by adding both terms.

**Effect of nearest neighbors and RoI pairs.** Table 5b ablates the effect of the number of nearest neighbors $K$ and RoI pairs $N$ used for generating inter-RoI pairs in Stage 2 of ORL. We set $N = 10\%$ when ablating $K$, and set $K = 10$ when ablating $N$. We observe that retrieving more nearest neighbors leads to better performance since more nearest neighbors provide more diverse image-level pairs to the subsequent generation of inter-RoI pairs. Although setting $K = 20$ produces a slightly better performance, we choose $K = 10$ by default as a trade-off considering the tendency of the saturated performance. Our method is more robust to the number of retrieved top-ranked RoI pairs after image-level nearest-neighbor retrieval, with $N = 10\%$ performing slightly better.

**Comparison with multi-crop BYOL.** Prior work [4] has indicated that cropping multiple views of the same image can improve the performance of self-supervised learning methods pre-trained on ImageNet. To investigate whether our gains are due to more accurate object-instance comparison or simply more number of mixed views, we randomly crop four additional smaller views for BYOL to ensure the number and size of the input patches are equal to ORL (*i.e.*, $2 \times 224 + 4 \times 96$). As shown in Table 5c, different from the observation on ImageNet, simply adding more low-resolution crops tends to hurt the performance since it will further intensify the inconsistent noise on scene images. In contrast, ORL substantially outperforms this multi-crop variant, validating that the gains are truly due to our object-level representation learning mechanism.

**Comparison with ground truth bounding boxes.** In Stage 2, ORL requires an unsupervised region proposal algorithm to extract approximate object-based regions, which is inaccurate to some extent. We further investigate whether the performance can be improved when more accurate object regions are available, *i.e.*, with bounding box annotations. To this end, we replace our selective-search generated object proposals with ground truth bounding boxes provided from COCO `train2017` set, while keeping all other procedures unchanged. As shown in Table 5d, adopting ground truth (GT) bounding boxes performs inferior to selective search (SS). This is mainly due to that although the ground truth bounding boxes can provide more accurate object location, their numbers are too scarce compared with a large amount of region proposals generated by selective search. The more diverse

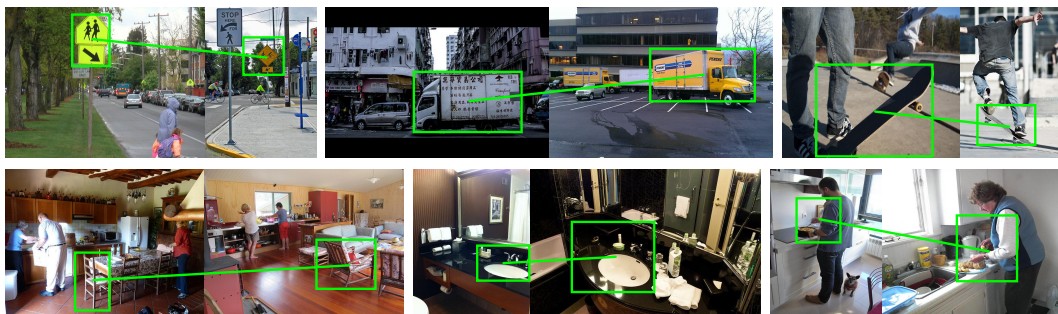

Figure 3: **Top-ranked region correspondence discovered by ORL in Stage 2.** We show a pair of discovered object-instance per image pair for clarity. More discovered correspondence pairs are provided in the supplementary material.

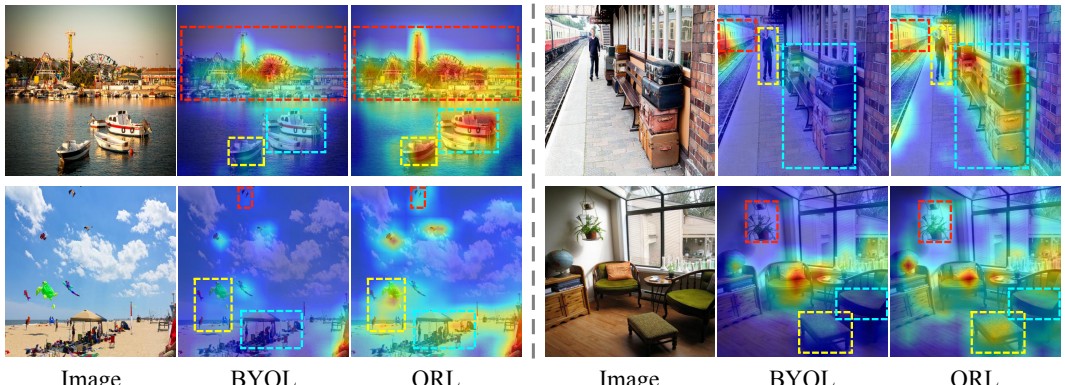

| Image | BYOL | ORL | Image | BYOL | ORL |

Figure 4: **Attention maps generated by BYOL and ORL.** ORL can activate more object regions and produce more accurate object boundary in the heatmap than BYOL. We provide more attention maps in the supplementary material. Best viewed with zoom in.

region proposals can potentially induce more unknown object or object-part discovery beyond the manually annotated objects. In this case, the diversity can make up for the inaccuracy.

**Pre-training schedules.** Table 5e shows the results with different pre-training schedules, from 100 epochs to 1600 epochs. The performance of both ORL and BYOL improves when pre-trained for longer epochs, while ORL consistently outperforms BYOL by at least 2.2% mAP. Note that our 200-epoch ORL has already surpassed the 1600-epoch BYOL (85.2% mAP vs. 84.9% mAP), demonstrating that the performance efficiency of ORL is at least $8\times$ than BYOL below 1600 epochs.

### 4.3 Visualization

**Correspondence pairs.** Figure 3 visualizes some top-ranked region correspondence discovered in Stage 2 of ORL. We observe that each generated inter-RoI pair largely correspond to the regions with similar visual concepts (*i.e.*, objects or object parts) across images. In contrast to typical contrastive learning methods that perform aggressive intra-image augmentations to simulate intra-class variances, our discovered inter-RoI pairs can substantially provide more natural and diverse intra-class variances at the object-instance level.

**Attention maps.** Figure 4 visualizes the attention maps generated by BYOL and ORL. We observe that both BYOL and ORL can produce relatively high-quality attention maps that focus on the foreground objects. This reflects from the side that current image-level contrastive learning methods have already induced a latent space with rich visual concepts. Nevertheless, ORL can activate more object regions and produce more accurate object boundary than BYOL in the generated attention maps. It is mainly due to introducing object-level similarity learning into ORL, which can minimize

the inconsistent noise caused by image-level contrastive learning. In contrast, BYOL only uses the whole image to extract features, thus activating the most discriminative region.

## 5  Conclusion

In this work, we have presented a new self-supervised learning framework, ORL, for object-level representation learning from scene images. We leverage the latent prior of image-level self-supervised pre-training for discovering object-based region correspondence across images. The generated object-instance correspondence enables us to perform pairwise contrastive learning at the object level. ORL significantly improves the performance of self-supervised learning from scene images in a variety of downstream tasks. We expect that our method can be applied to larger-scale unlabeled data in the wild to fully realize its potential, and hope that our study can attract the community's attention to more general-purpose unsupervised representation learning from scene images.

## Limitations

In this paper, we mainly perform pre-training experiments with ResNet-50 on COCO dataset, and further scale them up on COCO+ dataset. However, the promise of self-supervised learning is to harness massive unlabeled data by scaling up to ever-larger datasets. Some prior works [18, 3, 16] have attempted to leverage larger models and datasets to explore the limit of current self-supervised learning methods. For instance, a recent representative work SEER [16] performs billion-scale self-supervised pre-training on internet images using the RegNet architectures [46] with 700M parameters over 512 GPUs. Training at scale requires huge computational resources that are inaccessible to many researchers, which is not the core of our paper. We wish to highlight that our general-purpose ORL has yielded better performance than concurrent works [58, 34] that are tailored for dense prediction downstream tasks when pre-trained on COCO (Table 4), even surpassing the supervised ImageNet pre-training on several downstream tasks (Table 1-4). We expect that scaling ORL with larger architectures and datasets can further unleash its potential. Besides, ORL may not handle well on images with cluttered backgrounds since they will deviate the generated proposals to focus on these background regions. A possible remedy is to use some heuristic algorithms like saliency estimation to avoid the background regions. Another limitation is that ORL is a multi-stage framework. We expect an end-to-end framework to further improve the efficiency. We leave these explorations to future work.

## Broader impact

We present a more effective approach for learning unsupervised visual representations. Compared to supervised learning, it can liberate humans from expensive annotations as well as take advantages of rapidly growing real-world data. Like other learning algorithms, self-supervised learning should be applied with cautions when deployed in the real-world scenario. First, it is susceptible to biased learning if the algorithm is given with biased data. The exposure to unlabeled data may amplify such biases. Thus, debiasing measures have to be taken. Second, it remains non-trivial to dissect what is learned by self-supervised models. Similar concerns about the calibration, robustness, and interpretability of supervised models are equally applicable to the unsupervised counterpart. Our work is limited to the improvement of self-supervised learning within our scope. However, we acknowledge the importance of providing more transparent explanations for classification decisions, as well as the credibility of each prediction. Finally, our method still relies on the traditional regime of centralized learning. Privacy can be compromised if the method is applied on an unsaved platform. Federated learning can be a solution. How to scale self-supervised learning to the regime of decentralized learning will be an interesting research question to answer.

## Acknowledgements

This study is supported under the RIE2020 Industry Alignment Fund – Industry Collaboration Projects (IAF-ICP) Funding Initiative, as well as cash and in-kind contribution from the industry partner(s). The project is also supported by Singapore MOE AcRF Tier 2 (T2EP20120-0005), the Data Science and Artificial Intelligence Research Center at NTU.

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
