# Unsupervised Object-Level Representation Learning from Scene Images Supplementary Material

**Jiahao Xie**[1]   **Xiaohang Zhan**[2]   **Ziwei Liu**[1]   **Yew Soon Ong**[1,3]   **Chen Change Loy**[1]

[1]Nanyang Technological University
[2]The Chinese University of Hong Kong
[3]A*STAR, Singapore
{jiahao003, ziwei.liu, asysong, ccloy}@ntu.edu.sg
xiaohangzhan@outlook.com

## A   Training details for transfer learning experiments

**Image classification with linear models.** We follow the procedure in [4, 9] to train linear models on the frozen representations of ResNet-50. On VOC07, we train linear SVMs using the LIBLINEAR [3] package on the `trainval` split and report mAP on the `test` split. For linear evaluation on other datasets (ImageNet, Places205, and iNaturalist18), we train linear models with SGD using a batch size of 256, a learning rate of 0.01 decayed by 0.1 at three equally spaced intervals, a momentum of 0.9, and a weight decay of 0.0001. We train the linear models for 28 epochs on ImageNet, 14 epochs on Places205 and 84 epochs on iNaturalist18 to keep the number of training iterations roughly constant across all three datasets. We report top-1 center-crop accuracy on the validation split of ImageNet, Places205 and iNaturalist18.

**Low-shot image classification.** We train linear SVMs on the frozen representations of ResNet-50 following the same procedure in [4]. We train on the `trainval` split of VOC07 and report mAP on the `test` split. The results are averaged across five independent runs.

**Semi-supervised learning.** We use the 1% and 10% ImageNet subsets specified in the official code release of SimCLR [1]. We fine-tune the pre-trained ResNet-50 models with SGD for 20 epochs, using a batch size of 256, and a momentum of 0.9. We set the initial learning rate of backbone as 0.01 and that of linear classifier as 1. The learning rate is decayed by 0.2 at 12 and 16 epochs. We use a weight decay of 0 for 1% fine-tuning and 0.0001 for 10% fine-tuning.

**Object detection and segmentation.** We use a Mask R-CNN model [6] with R50-FPN backbone [7] implemented in Detectron2 [12], following the same fine-tuning setup in [10]. Specifically, we use a batch size of 2 images per GPU (a total of 8 GPUs), and fine-tune the ResNet-50 models for 90k iterations (standard 1× schedule). The learning rate is initialized as 0.02 with a linear warmup for 1000 iterations, and decayed by 0.1 at 60k and 80k iterations. The image scale is $[640, 800]$ pixels during training and 800 at inference.

## B   Reproducing related methods

We re-implement each image-level contrastive learning method on COCO as faithfully as possible. The implementation details of our most essential image-level baseline, *i.e.*, BYOL [5], are provided in Section 3.2 of the main text (the similar pre-training settings are also adopted in [8] while our reproduction achieves *better* results as shown in Table 1). For MoCo v2 [2], we follow [11] to set the initial learning rate as 0.3. Other hyper-parameters use the default values in the original paper. For SimCLR [1], we use the largest batch size (*i.e.*, 512) that fits in 8-GPU memory and linearly scale the

35th Conference on Neural Information Processing Systems (NeurIPS 2021).

Table 1: **Our reproduced results vs. existing results for BYOL.** All are based on 800-epoch pre-training on COCO with ResNet-50. The results are object detection fine-tuned on COCO using Mask R-CNN R50-FPN with default $1\times$ schedule.

| Method | Pre-train data | $AP^{bb}$ | $AP^{bb}_{50}$ | $AP^{bb}_{75}$ |
|---|---|---|---|---|
| BYOL repro. in [8] | COCO | 38.5 | 58.9 | 41.7 |
| BYOL repro. by us | COCO | 39.5 | 59.3 | 43.2 |

learning rate with the batch size (*i.e.*, $lr = 0.3 \times \text{BatchSize}/256$). Other hyper-parameters exactly follow the original paper.

## C  Additional visualization

**Correspondence pairs.** Figure 1 visualizes more top-ranked region correspondence pairs discovered by ORL in Stage 2. The high-response region pairs are usually objects or object parts.

**Attention maps.** Figure 2 visualizes more attention maps generated by BYOL and ORL. As analyzed in Section 4.3 of the main text, although image-level BYOL can generate relatively high-quality attention maps focusing on some of the objects in scene images, our object-level ORL can activate more regions that contain objects and produce more accurate object boundary in the heatmap than BYOL.

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

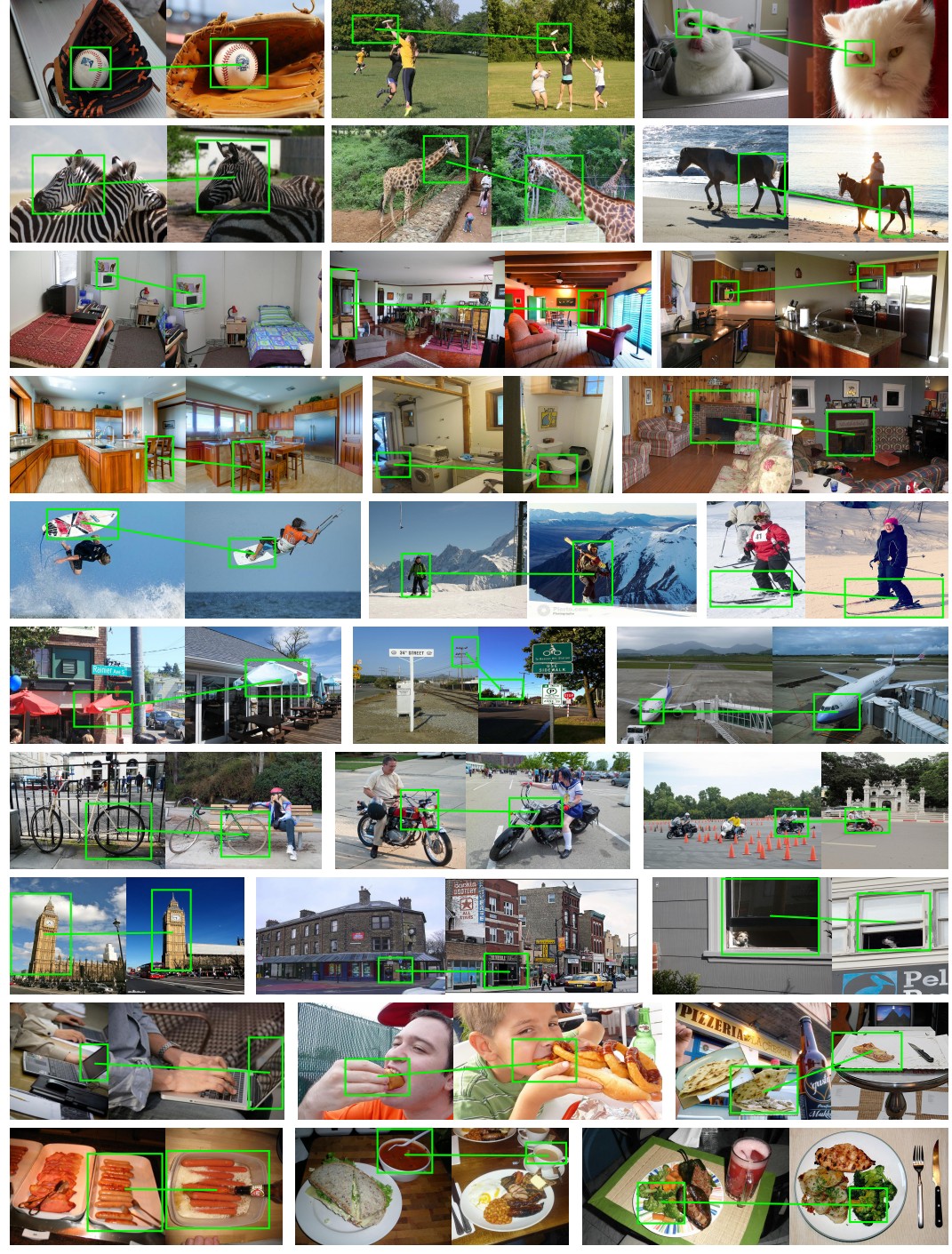

Figure 1: **Top-ranked region correspondence discovered by ORL in Stage 2.** We show one discovered object-instance pair per image pair for clarity.

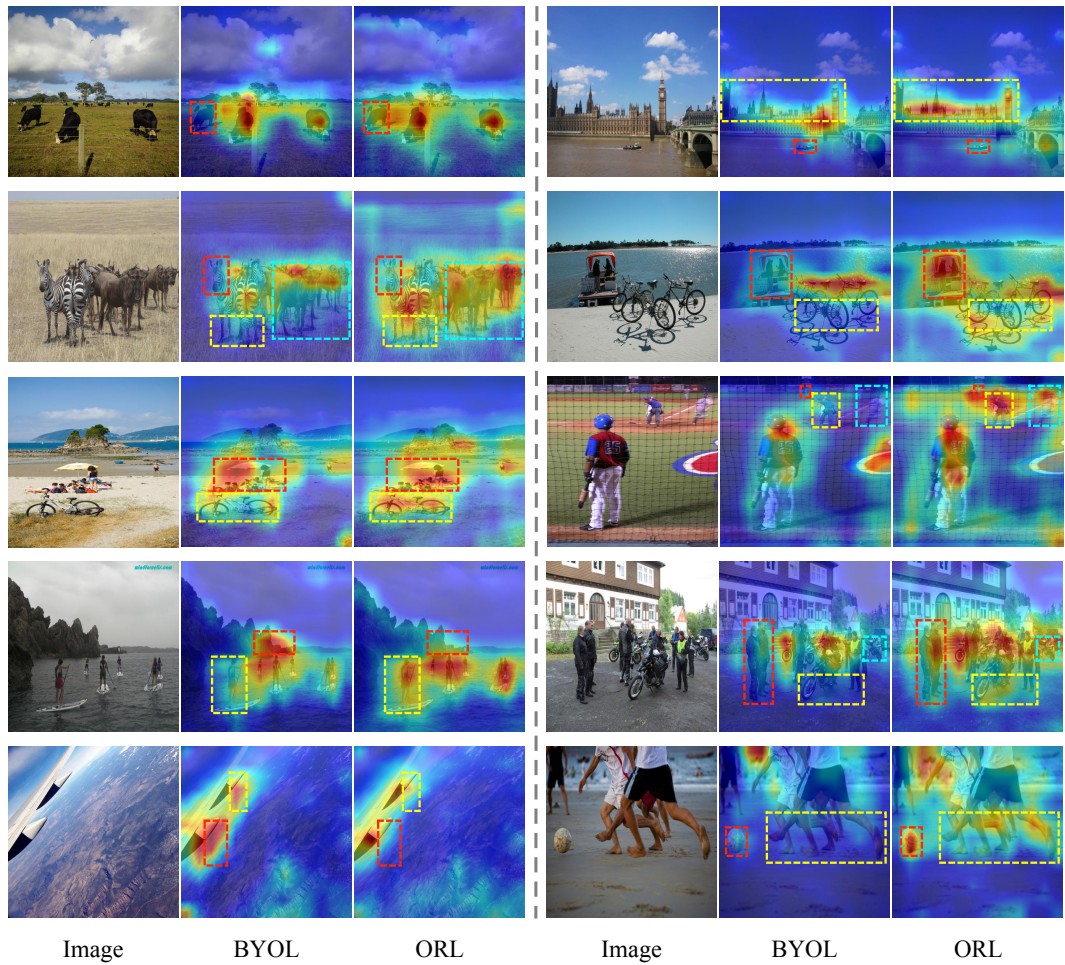

Image        BYOL        ORL        Image        BYOL        ORL

Figure 2: **Attention maps generated by BYOL and ORL.** ORL can activate more object regions and produce more accurate object boundary in the heatmap than BYOL. Best viewed with zoom in.