# OpenReview forum: "Unsupervised Object-Level Representation Learning from Scene Images"
_NeurIPS.cc/2021/Conference — NeurIPS 2021 Poster_

### Official Review · Reviewer_fNXx · 2021-07-15

**Rating:** 6
**Confidence:** 4

**Summary:**

This paper aims to learn object-centric self-supervised representation from non-iconic scene images. In addition to the standard self-supervised objective (e.g., BYOL), the paper suggests attracting the object patch with the (i) nearby patch of the chosen object and (ii) similar objects in different images. For (ii), the paper uses the similarity score of the standard image-level self-supervised model. This additional training signal from similar object patches improves the transfer performance of the learned representation on classification and detection/segmentation tasks.

**Limitations And Societal Impact:**

Both limitations and societal impact are well discussed in Appendix, though they could be moved to the main text following the intention of extending the page limits to nine pages.

**Main Review:**

Learning representations from non-iconic images is an important direction of self-supervised learning. There have been a large number of concurrent works on this topic (including DenseCL and Self-EMD discussed in the paper), but most of them reported the trade-off between image-level (e.g., classification) and pixel/object-level (e.g., detection/segmentation) tasks. Unlike the prior work that leverages the spatial features before the final pooling layer to improve detection performance, this paper simply matches the pooled features of related object patches, improving both classification and detection performance. Furthermore, the paper demonstrates that the model can find the visual correspondence of objects and produces a better attention map.

While the paper is well-written and easy-to-follow, I have two questions in detail:

I. How does the three-stage pipeline is exactly implemented?

- How long does the image-level model in Stage 1 pretrained? Is it a strict two-step approach of training Stage 1 and Stage 3 for 800 epochs, respectively? Does Stage 3 reinitialize the model or fine-tune the one from Stage 1? If latter, the other baseline models also should be fine-tuned again using the same computational budget.
- How often does Stage 2 perform? Are the correspondence maps only computed once? Do the correspondence maps become better when using the model trained from Stage 3? If it gives a better correspondence, would additional repetitions of Stage 2 and 3 give further improvement?

II. How does the attention maps are produced?

- The results seem promising, but I could not find the implementation detail. Also, it would be informative to provide quantitative results.

p.s. Typo: Missing punctuation after textbf in line 203.

**Time Spent Reviewing:**

4

---

> ### Author Response · Authors · 2021-08-10
> **Response from the Authors**
>
> Thank you for the constructive comments. Please find the following for our response.
>
> **Q1: How is the three-stage pipeline exactly implemented?**
>
> **A1:** (1) The models in Stage 1 and Stage 3 are all trained from scratch for 800 epochs, respectively.
> (2) The correspondence in Stage 2 is only computed once. We do not repeat Stage 2 and Stage 3 considering the trade-off between performance and computational budget. We agree that repeating Stage 2 and Stage 3 may give further improvements but is not the core of this paper. We leave these “self-iterative” explorations to future work.
>
> **Q2: How are the attention maps produced?**
>
> **A2:** We visualize the attention maps from Res5 stage of ResNet-50 after pre-training on the COCO dataset. To get relatively clear attention maps, we first enlarge the input size from 3x224x224 to 3x448x448. The shape of output tensor from Res5 is 2048x14x14. We then calculate the mean of tensor along the channel dimension and normalize the value to 0-1 to get the attention map, whose shape is 1x14x14. We further upsample the attention map to the input image size using bilinear interpolation and project the attention map onto the image to get the final visualized results. For the quantitative results, we follow the common evaluation protocol to transfer the learned features to various downstream tasks. The results have been provided in Table 1-5 of the paper.
>
> **Q3: Typo in L203.**
>
> **A3:** We will correct the typo in the final version.
>
> **Q4: Discussion on limitations and societal impact should be moved to the main text.**
>
> **A4:** We will move the discussion in the main text as suggested.

---

> > ### Comment · Reviewer_fNXx · 2021-08-13
> > **Response to the Rebuttal**
> >
> > Thanks for the response. I sincerely read all reviews and authors' rebuttal.
> >
> > I agree with Reviewer 5LqK and Reviewer k49a that the technical contribution is not very significant, and the proposed solution is more like engineering. However, I **keep my original rating of 6** since the paper is technically sound, and object-level learning is an important direction of visual self-supervised representation learning.

---

### Official Review · Reviewer_k49a · 2021-07-15

**Rating:** 6
**Confidence:** 4

**Summary:**

This work proposes a technique to adapt self-supervised contrastive learning to scene images where multiple objects are present in each image. Specifically, the proposed technique has three stages: 1. image-level pre-training with standard contrastive learning; 2. Finding object pairs images (crops) using selective search proposals and pre-trained features from stage-1; and 3. Doing contrastive learning with object image pairs. Several experimental comparisons demonstrate consistent improvements w.r.t. baselines that does only image-level contrastive learning.

**Limitations And Societal Impact:**

Limitations and broader impact are added in the supplementary. It is better to make the discussion of limitations more specific to this work (like, when the proposed algorithm fails and possible remedies) instead of mentioning general limitations of learning on large-scale datasets.

**Main Review:**

Strengths:
- Using self-supervised image features and selective search for correspondence object discovery is simple yet interesting technique.
- Experimental results on wide range of datasets and on several tasks.
- Consistent performance improvements w.r.t. image-level self-supervised learning techniques.

Weaknesses:
- The proposed stage-wise technique is more like an engineering solution to mine more/cleaner positive and negative pairs for contrastive learning. As such, the technical innovation is incremental.
- There is no discussion or comparisons with several works that extend image-level self-supervised learning to more local pixel features. Couple of works that I can find: [a,b].
- L65 says that this is the first study to extent self-supervised learning beyond image-level contrastive learning. This is an over-claim and not correct in my opinions as there are several existing works that propose pixel-level contrastive learning techniques.

Clarifications:
- Line 51 says that the proposed technique uses 'off-the-shelf contrastive learning pre-trained model'. Does this mean the stage-1 is already pre-trained on ImageNet data? Or, is it trained from scratch on COCO dataset? This point is important to clarify in the rebuttal.

[a] Pinheiro et al. Unsupervised learning of dense visual representations. NeurIPS 2020.
[b] Chaitanya et al. Contrastive learning of global and local features for medical image segmentation with limited annotations. NeurIPS 2020.

**Post Rebuttal**:

It seems reviewers generally agree that the proposed approach is more like an engineered system rather than providing any sound technical innovations. Like other positive reviewers, I do not find any clear issues in this work and experiments are reasonably convincing. It would be better if authors had provided experimental comparisons to more dense pixel-level contrastive learning works. After carefully considering author response and reviewer discussion, I tend to keep my original rating (6).

**Time Spent Reviewing:**

2

---

> ### Author Response · Authors · 2021-08-10
> **Response from the Authors**
>
> Thank you for the constructive comments. Please find the following for our response.
>
> **Q1: ORL is more like an engineering solution to mine more/cleaner positive and negative pairs for contrastive learning.**
>
> **A1:** We would like to clarify that constructing positive/negative pairs on scene images is non-trivial when no object annotations are available. The inconsistent learning signal caused by random cropping on scene images is the reason why we want to mine cleaner object-instance pairs. Table 5c in the paper has shown that mining cleaner pairs is crucial to improve representation learning on scene images. More importantly, we would like to highlight that this is the first work that proposes the explicit “object” notion and leverages the object-level information for general-purpose (not task-specific) representation learning on scene images. We think that our work presents important insights for advancing the self-supervised learning field towards scene images and can serve as a solid foundation for future research.
>
> **Q2: Discussion or comparisons with pixel-level contrastive learning works.**
>
> **A2:** Several pixel-level contrastive learning works have been discussed in L93-98. The performance comparisons with state-of-the-art pixel-level contrastive learning methods have been provided in Table 4. Our general-purpose ORL has substantially outperformed the concurrent works like DenseCL and Self-EMD that are specifically designed for dense prediction downstream tasks. Here, we would like to further discuss the differences of our work with [a, b]. For [a], (1) it adopts geometric transformation to induce low-level pixel correspondence within the same image to construct positive pairs, while ours leverages image-level prior to induce high-level object semantic correspondence across images to construct positive pairs; (2) it is specifically designed for dense prediction downstream tasks, while ours targets at more general-purpose representation learning that improves performance in both dense prediction and classification tasks. For [b], the pre-training is specifically designed for segmentation of volumetric medical images in the semi-supervised setting by leveraging domain-specific and problem-specific cues, while we focus on generic unsupervised pre-training from scene images. We will add the above discussion to our paper.
>
> **Q3: Over-claim in L65.**
>
> **A3:** Thanks for pointing out. We will change this claim to “We contribute the first study for object-level self-supervised learning”.
>
> **Q4: Clarify the pre-trained model in Stage 1.**
>
> **A4:** It is trained from scratch on COCO dataset.
>
> **Q5: It is better to make the discussion of limitations more specific.**
>
> **A5:** Some possible specific limitations are: (1) ORL may not handle well on images with cluttered backgrounds since they will deviate the generated proposals to focus on these background regions. A possible remedy is to use some heuristic algorithms like saliency estimation to avoid the background regions. (2) Another limitation is that ORL is a multi-stage framework. We expect an end-to-end framework and leave these explorations for future works. Thank you for the suggestion and we will add more discussion in the final version.

---

### Official Review · Reviewer_JkCU · 2021-07-16

**Rating:** 4
**Confidence:** 4

**Summary:**

This paper proposes an object-level self-supervised pre-training approach which extends and tries to improve existing work on image-level self-supervised pre-training. The authors motivate the problem by mentioning that image-level self-supervised pre-training methods are infeasible for images with many objects due to the hard constraints imposed by such methods. The paper proposes to apply self-supervised pre-training to similar objects in and across images. The authors use pre-trained image-level model to select the best candidate object pairs across images and then pre-train self-supervised models using the selected object regions. The authors report some performance improvements on the COCO dataset.

**Limitations And Societal Impact:**

The authors have adequately discussed the limitations and societal impact, but only in the supplementary material and not as part of the text of the main paper.

**Main Review:**

The paper proposes a slightly interesting approach for self-supervised pre-training. The proposed approach achieves promising initial results - improving over existing image-level pre-training approaches and even fully-supervised baselines. The motivation behind this work is sound. and the authors have explained the limitations of existing works well.

However, a large number of aspects have not been considered in the paper. Such aspects need to be dealt with to ensure completeness of the paper. I list a few major limitations of the paper here.

1. The most important limitation of the paper is a missing study on how the proposed approach scales with the availability of more data. The authors should try to train their model on larger datasets like OpenImages. This will show whether the proposed approach has any potential for larger-scale training.

2. The use of a image-level model to first find the nearest images/objects and then training the same model on the retrieved objects seems circular. It's not clear why the model should change/learn anything new if the similar objects are already retrieved using the same model. What additional information does this framework gain?

3. The authors should also provide a baseline by training the ORL on the ImageNet dataset to highlight the problem with using iconic images for pre-training.

4.  In table 5d, the authors compare the performance when using selective search vs the performance when using ground-truth boxes. The authors show that using the ground-truth boxes leads to a lower performance than when using selective search. The authors hypothesize that this might be due to more diverse boxes returned by selective search. Did the authors try using jittered ground-truth boxes to induce more diversity in the ground-truth boxes? That might lead to better performance than selective search.

5. Finally, I think the paper can be made much stronger if the authors conduct downstream experiments on datasets other than COCO. Does the proposed approach lead to similar performance improvements when the pre-training dataset is different from the downstream dataset?

**Time Spent Reviewing:**

4

---

> ### Author Response · Authors · 2021-08-10
> **Response from the Authors**
>
> Thank you for the constructive comments. Please find the following for our response.
>
> **Q1: Scalability on larger datasets.**
>
> **A1:** We have demonstrated the scalability of our method on a larger “COCO+” dataset in the paper. As stated in the Limitation Section of the supplementary material, training at scale like OpenImages requires huge computational resources that are inaccessible to many researchers, which is not the core of this paper. It will be unfair if some works are penalized for missing such commonly unaffordable larger-scale experiments. We acknowledge that training on even larger datasets are interesting experiments to explore but leave these larger-scale studies for future works.
>
> **Q2: What additional information does the framework gain?**
>
> **A2:** The framework benefits from more fine-grained object-level information. Prior works do not fully exploit the object-level information residing in scene images. We explicitly model this object-level information as additional training signals. The improvements on various downstream tasks, the ablations and the visualizations have all demonstrated the advantages of leveraging this fine-grained information.
>
> **Q3: A baseline to pre-train ORL on ImageNet.**
>
> **A3:** We would like to clarify that, since we want to leverage scene images, the premise of ORL is that there should be multiple objects per image. The techniques presented in the paper (e.g., object proposal generation, multi-object correspondence discovery) are specifically designed for this scenario. It is apparently unnecessary to apply these techniques on ImageNet with an average of only 1.1 objects per image. We have highlighted the problem with using iconic images for pre-training in Section 1. Extensive experiments in Section 4 have demonstrated that ORL can indeed improve the performance of image-level self-supervised learning on scene images.
>
> **Q4: Justification in Table 5d.**
>
> **A4:** As stated in L272-274, we only replace the selective-search bounding boxes with the ground-truth ones while keeping all other procedures including jittering bounding boxes unchanged. Thus, we have already tried jittering the ground-truth boxes. However, the performance is still worse than its selective-search counterpart. We expect that a better performance could be achieved if the ground-truth cross-image correspondence is readily available. However, it is very expensive to annotate such cross-image correspondence. This demonstrates the superiority of unsupervised learning over supervised learning since class labels are actually too sparse.
>
> **Q5: Downstream experiments on datasets other than COCO.**
>
> **A5:** We have conducted various non-COCO downstream experiments as shown in Table 1-3 of the paper. ORL still exhibits significant performance improvements on these non-COCO downstream tasks.
>
> **Q6: Limitations and societal impact sections are not put in the main text.**
>
> **A6:** We will put these sections in the main paper as suggested.

---

> ### Author Response · Authors · 2021-08-24
> **Looking forward to the feedback**
>
> We have clarified the listed concerns in detail. We would love to hear your feedback on whether our answers have solved your concerns or you have further questions.

---

### Official Review · Reviewer_5LqK · 2021-07-19

**Rating:** 7
**Confidence:** 4

**Summary:**

This paper presents a new self-supervised learning framework that leverages scene images to enhance object-level representation learning that only relies on object images. The basic idea is to apply unsupervised region proposal and k-NN based object similarity to diversify intra- and inter-image pairs with more variances to aid object-level representation learning. The improvements on various downstream tasks, the ablations and visualizations all demonstrate the advantages of the presented framework. The main contribution is to utilize scene images for contrastive self-supervised learning.

**Limitations And Societal Impact:**

The limitations and broader impact of the work are discussed in the supplementary material and look good to me.

**Main Review:**

The main novelty of the paper is to enhance the contrastive self-supervised learning by including not only object images from ImageNet, but also scene images from Coco. The paper is written well in general and the experiments are extensive.

I did not give a higher rating due to the following:
- The technical contribution of the paper is not very significant as the multi-stage framework largely depends on previous contrastive learning work as BYOL. The improvement is basically due to the data augmentation from Coco which is not technically difficult.
- The generality of the framework could have been better demonstrated by not just on BYOL. Please discuss other possibilities in more detail.
- The trade-off between the performance gain and the additional cost due to multiple stages needs to be discussed more.

Some minor things of writing:
- Please cite the published paper other than its preprints on arXiv (e.g., BYOL)
- L169 the symmetric loss is not clear here.
- L104 ‘detect’ -> ‘detecting’
- L129 ‘is comprised of’ -> ‘comprises’
- L203 ‘models Following’ -> ‘models. Following’


**Time Spent Reviewing:**

2

---

> ### Author Response · Authors · 2021-08-10
> **Response from the Authors**
>
> Thank you for the constructive comments. Please find the following for our response.
>
> **Q1: The technical contribution is not very significant as it depends on BYOL. The improvement is basically due to the data augmentation which is not technically difficult.**
>
> **A1:** To begin with, we would like to highlight that we contribute the first study for object-level self-supervised learning (SSL). The promise of SSL methods is that they ought to allow us to learn better features from unconstrained scene images in the wild rather than from highly curated object-centric images. Therefore, it is imperative to evolve SSL from image level to object level. Compared with a large body of existing SSL works that focus on ImageNet, our work proposes the explicit “object” notion in SSL for the first time and presents important insights for advancing the field towards scene images. Our first attempt w.r.t. object-level SSL can serve as a good foundation for future development.
>
> (1) Our framework is based on two interesting observations that have not been explored before for SSL:
> (i) image-level contrastive learning encodes priors for region correspondence discovery across images, and
> (ii) high-response regions are usually objects or object parts.
> These two observations are general and orthogonal to the architectures we use to instantiate our idea. We adopt BYOL in the paper due to its state-of-the-art image-level performance. It should serve as a very strong baseline on which we can demonstrate the effectiveness of our approach. Certainly, BYOL can be replaced with other image-level contrastive learning methods as discussed in **A2**.
>
> (2) We would like to clarify that our work is not a simple data-augmentation strategy. A series of technical designs (e.g., object proposal generation, multi-object correspondence discovery) all aim at better exploiting object-level information naturally residing in scene images. Naïvely adopting low-level data augmentations cannot obtain such fine-grained high-level semantic information.
>
> **Q2: Discussion on the generality of the framework.**
>
> **A2:** We adapt MoCov2 to our framework as an example to demonstrate the generality. The results based on 800-epoch pre-training on COCO with ResNet-50 are listed below (the low-shot results are averaged across all low-shot values):
>
> | Method     | VOC07 | ImageNet | Places205 | iNat. | VOC07 low-shot | ImageNet 1% | ImageNet 10% |
> |------------|:-----:|:--------:|:---------:|:-----:|:--------------:|:-----------:|:------------:|
> |            |  **mAP**  |   **Top-1**  |   **Top-1**   | **Top-1** |       **mAP**      |    **Top-1**    |     **Top-1**    |
> | MoCov2     |  82.2 |   55.1   |    48.8   |  27.8 |      66.1      |     28.2    |     57.1     |
> | MoCov2-ORL |  86.4 |   58.6   |    52.3   |  31.2 |      69.3      |     30.6    |     59.8     |
>
> Our MoCov2 variant of ORL (i.e., MoCov2-ORL) outperforms the MoCov2 baseline on various downstream tasks by significant margins, demonstrating that our general framework is not confined to specific contrastive learning methods. We will show more variants in the final version.
>
> **Q3: Discussion on the trade-off between the performance gain and the additional cost.**
>
> **A3:** Compared with end-to-end methods, our multi-stage framework requires an additional image-level pre-training stage. Thus, the computational budget of ORL is 2x than BYOL. We trained BYOL for a longer 2x schedule (i.e., 1600 epochs) and its performance on VOC07 SVM classification benchmark is 84.9% mAP. Our 800-epoch ORL (86.7% mAP) still substantially outperforms 1600-epoch BYOL (84.9% mAP) by 1.8% mAP. Besides, as shown in Table 5e, our 200-epoch ORL (85.2% mAP) has already surpassed the 800-epoch BYOL (84.5% mAP), even outperforming the 1600-epoch BYOL (84.9% mAP). This indicates that the performance efficiency of ORL is at least 8x than BYOL below 1600 epochs, which can obviously compensate for its additional 2x computational budget.
>
> **Q4: Minor things of writing.**
>
> **A4:** In L169, the symmetric loss is computed by separately feeding $v’$, $p’$, $p_2$ to the online network and $v$, $p$, $p_1$ to the target network. We will make this point clearer and fix other issues in the final version.

---

> > ### Comment · Reviewer_5LqK · 2021-08-14
> > **Response to the rebuttal**
> >
> > Thanks for the detailed reply with additional results. The explanations are clear and the new evidence of generality is convincing. I have increased my rating to 7 to support the paper.

---

### Decision · Program_Chairs · 2021-09-27

**Decision:**

Accept (Poster)

**Comment:**

- The proposed method is tackling an important problem. The reviewers found the approach is reasonable but technically not very novel.
- The many concerns from the reviewers are addressed well enough by the rebuttal.
- The strength is mainly the experiment results which are wide, consistent and reasonably convincing.
- The clarity is good enough but can be improved.